Can 13C stable isotope analysis uncover essential amino acid provisioning by termite-associated gut microbes?

Ayayee Paul A. 1 akwettey@gmail.com
Jones Susan C. 2
Sabree Zakee L. 1
1 Department of Evolution, Ecology and Organismal Biology, The Ohio State University , Columbus, OH , USA
2 Department of Entomology, The Ohio State University , Columbus, OH , USA
Smidt Hauke
Electronic publication date: 2015 Aug 27
Publication date: 2015
Volume: 3
Electronic Location ID: e1218
Received 2015 May 21; Accepted 2015 Aug 7
Copyright: © 2015 Ayayee et al.
Copyright year: 2015
Copyright holder: Ayayee et al.
License: This is an open access article distributed under the terms of the Creative Commons Attribution License, which permits unrestricted use, distribution, reproduction and adaptation in any medium and for any purpose provided that it is properly attributed. For attribution, the original author(s), title, publication source (PeerJ) and either DOI or URL of the article must be cited.
License URL: https://creativecommons.org/licenses/by/4.0/

Keywords: Essential amino acid, Gut microbiome, Reticulitermes flavipes, 13C-stable isotope analysis, Symbiosis

Funding: The Ohio State University State and Federal funds The Ohio State University for supporting PAA and ZLS with funds for this study. Research support for SCJ was in part from State and Federal funds appropriated to the Ohio Agricultural Research and Development Center, The Ohio State University. The funders had no role in study design, data collection and analysis, decision to publish, or preparation of the manuscript.

==============================
Gut-associated microbes of insects are postulated to provide a variety of nutritional functions including provisioning essential amino acids (EAAs). Demonstrations of EAA provisioning in insect-gut microbial systems, nonetheless, are scant. In this study, we investigated whether the eastern subterranean termite Reticulitermes flavipes sourced EAAs from its gut-associated microbiota. δ13CEAA data from termite carcass, termite gut filtrate and dietary (wood) samples were determined following 13C stable isotope analysis. Termite carcass samples (−27.0 ± 0.4‰, mean ± s.e.) were significantly different from termite gut filtrate samples (−27.53 ± 0.5‰), but not the wood diet (−26.0 ± 0.5‰) (F(2,64) = 6, P < 0.0052). δ13CEAA-offsets between termite samples and diet suggested possible non-dietary EAA input. Predictive modeling identified gut-associated bacteria and fungi, respectively as potential major and minor sources of EAAs in both termite carcass and gut filtrate samples, based on δ13CEAA data of four and three EAAs from representative bacteria, fungi and plant data. The wood diet, however, was classified as fungal rather than plant in origin by the model. This is attributed to fungal infestation of the wood diet in the termite colony. This lowers the confidence with which gut microbes (bacteria and fungi) can be attributed with being the source of EAA input to the termite host. Despite this limitation, this study provides tentative data in support of hypothesized EAA provisioning by gut microbes, and also a baseline/framework upon which further work can be carried out to definitively verify this function.

Introduction

Associations between termites and their gut microbes are some of the most well-studied symbioses. The majority of lower termites (all families except Termitidae) are wood feeders that thrive on these nitrogen-limited diets (Mattson, 1980) by relying upon gut microbes that can fix atmospheric nitrogen (Lilburn et al., 2001; Meuti, Jones & Curtis, 2010). Termites are incapable of meeting nutritional demands for nitrogen-rich metabolites such as proteins, in the absence of these microbes. Termites also rely on gut microbes to metabolize plant tissues, comprised largely of cellulose, into assimilable carbon. Digestion of cellulose and hemicellulose is attributed to a consortium of termite, bacteria and protist-derived cellulases that ultimately liberate carbon in plant tissues (Scharf et al., 2011; Tartar et al., 2009; Warnecke et al., 2007). Evidence of 13C-metabolite transfer between protists and associated gut bacteria in the desert damp wood termite, Paraneotermes simplicicornis, further confirms the flow of nutrients in the termite gut following 13C-cellulose degradation by associated protists (Carpenter et al., 2013). An additional microbe-specific function that benefits the termite host is acetogenesis/carbon dioxide fixation (Breznak & Kane, 1990; Pester & Brune, 2006). Together, nitrogen fixation, and acetogenesis provide ammonia and acetate, respectively, which the host can use for biosynthetic and metabolic processes. Oxygen scavenging and removal of excess hydrogen via methanogenesis are additional microbe-specific functions that are essential to maintaining the physiological and biochemical conditions within the gut microenvironment, ensuring that the aforementioned processes can continue (Brune & Friedrich, 2000).

An important aspect of wood-feeding insects’ nutritional ecology is the acquisition of essential amino acid (EAAs) because these cannot be generated by the host de novo (Douglas, 2013). Proctodeal trophallaxis (mouth-anus transfer of gut contents among nestmates), an essential colony feature, is thought to serve as one of the means by which termites acquire EAAs (Nalepa, Bignell & Bandi, 2001). Briefly, partially digested and undigested materials (with dead and living microbial fractions) are ingested from the anus of colony members and are used for inoculation or digestion (Kitade, 2004). Inoculation is thought to be more relevant for newly eclosed (hatched) and molted (inter-stadial growth) colony members, and digestion the norm in workers (Kitade, 2004). Similarly, notably higher normalized intensities of 13C-labelled EAAs detected in the lumen fluids of the midgut relative to the foregut and hindgut at 24 h, following feeding on 13C-cellulose in the damp wood termites (Hodotermopsis sjostedti) is attributed to proctodeal trophallaxis and subsequent digestion of microbial fractions (Tokuda et al., 2014). It still remains to be determined conclusively, that termites acquire and assimilate EAAs from gut microbes, since only the gut lumen fluid, but not actual termite tissue were sampled.

In this study, we investigated the acquisition of EAAs from associated gut microbiota of the eastern subterranean termite, Reticulitermes flavipes, using naturally occurring variations in the 13C/12C ratios of EAAs from bacteria, fungi, and plants. The premise of this approach is two-fold. First, because insects are incapable of de novo EAA biosynthesis and rely solely on dietary sources, the 13C-signature (determined via isotope ratio mass spectrometry) of an EAA in an insect consumer (δ13CConsumer EAA) is expected to approximate that of the diet (δ13CDietary EAA), with little change in the ratio of 13C/12C stable isotopes in the carbon skeleton of that particular EAA (Caut, Angulo & Courchamp, 2008; McMahon et al., 2010; Newsome et al., 2011). The isotopic difference between consumer δ13CEAA and dietary δ13CEAA(given by the delta notation; Δδ13C = δ13CConsumer EAA − δ13CDietary EAA) is estimated to within 1‰ of the dietary δ13CEAA. Any significant deviation from the expected Δδ13C of 1 suggests the possibility of alternate or additional sources of EAAs (Newsome et al., 2011; Tieszen et al., 1983).

The second premise relies on the fact that plants, bacteria and fungi are the only organisms capable of synthesizing EAAs and non-essential amino acids de novo. Additionally, bacteria, fungi, and plants have unique and distinct EAA signatures as a result of different biosynthetic pathways and processes that eventually lead to different 13C/12C stable isotopes ratios. The distinct δ13CEAA signatures across these groups have been empirically demonstrated (Larsen et al., 2009) and used in several ecological studies (Larsen et al., 2011; Larsen et al., 2013; Vokhshoori, McCarthy & Larsen, 2014).

Thus, according to the first premise, a determined discrimination/offset factor (Δδ13C) greater than 1 between the δ13CEAA of a consumer and its diet, suggests the possibility of an additional/alternate source of EAAs for the consumer. The second premise enables the identification of the possible contributing source, within a predictive model framework, based on the unique δ13CEAA signatures of plants, bacteria, and fungi. It is important to stress that, the 13C-offset determination serves only to determine the use of non-dietary EAAs if any, and not to identify biosynthetic origin of such non-dietary EAAs.

In this study, we investigated the 13C-offset between termites and their dietary substrates and assessed biosynthetic contributions from plant, bacteria and fungi to termite δ13CEAA signature. Actual insect tissues in addition to gut lumen fluids were examined in order to conclusively determine the incorporation of microbial EAAs into the termite body.

Materials and Methods

Insects

R. flavipes originated from Orient, OH (39°46′19.99″N, 83°09′22.30″W) and were maintained in a laboratory colony in a plastic container fitted with a lid and provisioned with wood (a mixture of wood mulch and pine (Pinus spp.)), that was moistened periodically with distilled water. These colonies were maintained in the laboratory of Dr. Susan Jones. The termite colony originated from a single inbred colony that had been established during May 2010 by pairing a single male and female de-alate, and was maintained at room temperature (∼22 °C ± 2 °C) in the dark, under ambient laboratory conditions.

Sample collection and preparation

After 8 weeks of feeding, 50 individual workers were removed from the colony and surface sterilized by rinsing once in 10x Coverage Plus (Steris, Mentor, Ohio, USA) and twice in sterile distilled water. A total of 5 termite samples (n = 5, each made up of 10 pooled workers) were obtained. The entire alimentary system was removed from each worker and placed in a 1x phosphate buffered saline solution (PBS). The remaining termite carcass was place in a 1.5 ml Eppendorf tube (Eppendorf, Hauppauge, New York, USA). Hence, each termite sample was subdivided into the termite carcass (n = 5) and its gut fraction (n = 5). Pooled termite guts were homogenized in PBS and filtered through a 0.45 µm membrane filter (EMD Millipore, Billerica, Massachusetts, USA) to eliminate insect debris. Gut filtrates were stored at −80 °C for 48 h prior to lyophilization. Wood samples (n = 4) from the termite colony were also ground into a coarse powder in a coffee mill and frozen at −80 °C for 48 h before lyophilization. Termite and wood samples then were sent to the Stable Isotope Facility (SIF) at UC Davis, Davis, California, USA, for 13C-stable isotope analysis.

EAA stable isotope analysis

Freeze-dried termite and wood samples were acid hydrolyzed and derivatized resulting in the addition of a known carbon residue to the analytes of interest (Walsh, He & Yarnes, 2014). Non-analyte carbon correction was subsequently performed to correct for the addition of carbon during the derivatization process (Doherty, Jones & Evershed, 2001; Walsh, He & Yarnes, 2014). Approximately 0.2–0.5 µl aliquots of derivatized samples were injected into a splitless liner at 250 °C with a helium flow rate of 2.8 mL/min. Compound-specific isotope 13C-amino acid analysis (CSI- 13CAA) was performed using the TRACE GC Ultra gas chromatograph (GC; Thermo Fisher Scientific, Waltham, Massachusetts, USA) coupled to a Delta V Advantage isotope ratio mass spectrometer via the GC Combustion Interface III (Thermo Electron, Bremen, Germany) using the high polarity VF-23ms capillary column (Agilent Technologies, Santa Clara, California, USA). Combustion and reduction furnace temperatures were 950 °C and 650 °C, respectively. δ13C isotopic abundances are reported as δ13C values relative to the standard Vienna Pee Dee Belemnite (V-PDB) scale. δ13CEAA quantified from termite carcasses, termite gut filtrates, and wood samples, following a non-analyte correction relative to internal amino acid standards, and used in the statistical analyses.

Statistical analyses

Mixed model analysis and mean separations (Student’s t-test) were carried out on δ13CEAA data using JMP 10 (SAS Inc., North Carolina, USA). Overall 13C-offset between termite (δ13CTermite) and wood (δ13CWood) samples was determined as Δδ13C = (δ13CTermite–δ13CWood). Individual patterns of 13C-offset across EAAs between termite and wood samples was determined as Δδ13CEAA = (δ13CTermite EAA–δ13CWood EAA).

Calibration and model validation

An inter-lab calibration was performed to minimize instrumental error and/or variability between δ13CEAA data from our study and from Larsen et al. (2013) for representative fungi (n = 9), bacteria (n = 11), and plants (n = 12). The predictive model was validated using the reference bacteria, fungi and plant samples as classifiers to ascertain distinctness of each group. Additionally, δ13CEAA data obtained from the fungus Fusarium solani (n = 2), used in a previous study and analyzed at the same facility as these samples were, was used to further validate the separation of the classifiers by the model based on the particular EAAs used in the study.

This was followed by a supervised discriminant analysis to determine group membership of termite samples (carcass and gut filtrate) and wood samples to the respective classifier groups (Larsen et al., 2013). Classification of the training data and samples was performed using the jackknifed predictions. Linear discriminant function analysis (LDA) was carried out using the R package MASS (Venables & Ripley, 2002). We considered wood samples as predictors in the predictive modeling and not as classifiers.

Ethics statement

No animal rights were violated in the execution of this study and conditions were within the guidelines of the Ohio State University’s Office of Responsible Research Practices.

Results

The δ13C of all EAAs were quantified, and only those passing quality checking (having distinct and non-overlapping peaks obtained from the GC capillary column), were selected for further analysis. δ13CEAA data were obtained for isoleucine, leucine, valine, phenylalanine, lysine and threonine from all samples (Table 1). Complete δ13CEAA data, however, was available only for isoleucine, valine, phenylalanine and lysine across all samples. Leucine δ13C data was unavailable for three out of the five termite gut filtrate samples, due to failure to pass quality control (absence of distinct and non-overlapping peaks obtained from the GC capillary column). Similarly, Threonine δ13C data was unavailable for four out of five termite gut filtrate samples. Threonine was excluded from all further analyses. Leucine, however, was used in calculating the 13C-offset between termite samples and the wood diet, because there were two termite gut filtrate samples, for which this could be calculated relative to wood; but was omitted from the predictive model analysis due to the missing data points (Table S1).

Table 1 Summary 13C-data for all samples.

Mean δ13CEAA (mean of two technical replicates) of termite and wood samples following 13C-analysis at the UC Davis Stable Isotope Facility, showing data for all EAAs measured.

Samples	Ile	Leu	Lys	Phe	Thr	Val	
Termite carcass 1	−26.59	−29.69	−19.70	−26.84	−16.82	−27.40	
Termite carcass 2	−26.56	−29.85	−19.28	−26.54	−18.90	−26.79	
Termite carcass 3	−27.62	−31.20	−21.72	−27.25	−17.98	−28.92	
Termite carcass 4	−27.97	−31.51	−24.37	−27.66	−16.46	−28.77	
Termite carcass 5	−26.11	−29.72	−20.04	−26.25	−14.65	−26.89	
Termite gut filtrate 1	−42.35	N/A	−18.93	−25.37	N/A	−27.08	
Termite gut filtrate 2	−29.26	N/A	−20.80	−28.16	N/A	−27.68	
Termite gut filtrate 3	−31.30	−35.33	−18.05	−27.47	N/A	−28.79	
Termite gut filtrate 4	−32.96	N/A	−19.78	−27.40	N/A	−27.05	
Termite gut filtrate 5	−28.49	−32.16	−21.98	−27.92	−20.03	−27.41	
Wood 1A	−21.54	−26.27	−19.49	−26.35	−18.96	−24.83	
Wood 1B	−27.45	−30.26	−24.46	−29.78	−24.67	−29.53	
Wood 2B	−23.78	−29.66	−20.36	−27.44	−16.95	−27.74	
Notes.

N/A not available

δ13CEAA analysis summary and 13C-offset (Δδ13CEAA) between termite samples and wood diet

The overall model was significant (F(14,52) = 13.7, P < 0.0001, R-square = 0.80), with significant group (F(2,64) = 6.0, P < 0.0052) and amino acid (F(4,62) = 35, P < 0.0) effects, as well as significant group*amino acid interaction (F(8,58) = 5.1, P < 0.0001). Termite gut filtrate (−27.53 ± 0.5‰) (mean ± s.e) was significantly different from the wood diet (−26 ± 0.5‰) and the termite carcass (−27.0 ± 0.4‰)(Table 1A). Termite carcass and wood diet were not significantly different from each other. Termite carcass and termite gut filtrate samples were respectively −1.0 ± 0.4‰ and −2.3 ± 0.4‰ 13C-depleted relative to wood the diet (Table 2A).

Table 2 Summary statistics of between sample comparisons.

(A) Mean δ13CEAA and Δδ13C-offsets between termite samples (carcass and gut filtrate) and wood diet. Shown are mean values for termite (n = 5) and wood (n = 4) samples. Different letters represent a significant difference between groups (student’s t-test) (F(2,63) = 6.0, P < 0.004). (B) Group*amino acid interaction mean δ13CEAA values for each amino acids across all samples (F(8,58) = 5.1, P < 0.0001) (student’s t-test).

(A) Sample	δ13C data (Mean ± s.e ‰)	Δδ13C	
Termite carcass	−27.0 ± 0.4 (B)	−1.0	
Termite gut filtrate	−28.3 ± 0.5 (A)	−2.3	
Wood	−26.0 ± 0.5 (B)	0	
(F(2,64) = 6.0, P < 0.0052)			
(B) Amino acid*group	δ13C data (Mean ± s.e ‰)	
Ile, Termite gut filtrate	−32.87 ± 1 (E)	
Ile, Termite carcass	−26.97 ± 1 (C)	
Ile, Wood	−23.69 ± 1.1(B)	
Leu, Termite gut filtrate	−33.76 ± 1 (E)	
Leu, Termite carcass	−30.39 ± 1 (D, E)	
Leu, Wood	−29.03 ± 1.1 (D)	
Lys, Termite gut filtrate	−19.90 ± 1 (A)	
Lys, Termite carcass	−21.02 ± 1 (A, B)	
Lys, Wood	−22.22 ± 1.1 (A, B)	
Phe, Termite gut filtrate	−27.26 ± 1 (C)	
Phe, Termite carcass	−26.91 ± 1 (C)	
Phe, Wood	−27.63 ± 1.1 (C, D)	
Val, Termite gut filtrate	−27.60 ± 1 (C)	
Val, Termite carcass	−27.75 ± 1 (D)	
Val, Wood	−27.40 ± 1.1 (C)	
(F(8,58) = 5.1, P < 0.0001)		

The pattern of 13C-offset of the five-measured EAAs in the termite samples relative to the wood diet is presented in Fig. 1. The 13C-offsets presented in Fig. 1 are not intended to resolve the biosynthetic origins of EAAs, merely to determine if the calculated Δδ13CEAA between termite samples and wood provides an indication of the use of non-dietary EAAs (i.e., equal to or greater than 1‰). Briefly, phenylalanine, lysine and valine from the termite samples (carcass and gut filtrate) and wood diet were not significantly different from the each other; even though lysine in the termite samples was 13C-enriched relative to the lysine in the wood-diet, and phenylalanine and valine in termite samples were 13C-depleted, relative to the wood diet (Fig. 1) (Table 2B). Isoleucine and leucine from termite gut filtrate samples was significantly 13C-depleted relative to termite carcass and wood diet. Only isoleucine in termite carcass samples was significantly 13C-depleted relative to the wood diet (Fig. 1) (Table 2B).

Figure 1 δ13C-offset between termite samples and wood diet.

δ13C-offset (Δδ13CEAA) (enrichment or depletion) of 5 five essential amino acids (EAAs) in termite carcass (termites) and termite gut filtrate samples relative to the wood diet EAAs; Δδ13CEAA(‰) = (δ13CTermiteEAA–δ13CWood EAA). (F(2,63) = 6.2, P < 0.004). Shown are mean values for 5 replicates per termite sample (termite and termite gut filtrate) and 4 replicates for wood diet. The EAAs were isoleucine (Ile), lysine (Lys), phenylalanine (Phe), and valine (Val). Error bars represent standard errors of the mean.

Validation of predictive model and classification of termite sample EAAs

In the linear discriminant analysis (LDA) plots, the 95% confidence limits decision regions for each group/classifier are depicted as ellipses around the classifiers and the decision boundaries between the groups/classifiers as lines. After establishing the discrimination model, we then predicted posterior probabilities, i.e., the probability that a particular sample belonged to one or another of the three groups. The greater the distance of a particular consumer from the centroid of a classification group, i.e., potential EAA source, the greater the probability mixing of EAA sources occurred. Given the distinct discrimination scores between the classification groups, we interpreted discriminant scores of termite samples falling outside the 95% confidence limits of the food source, wood (plants) as strong indications of symbiotic EAA provisioning.

The predictive model was validated, based on the correct classification of bacteria (n = 11), fungi (n = 9) and plants (n = 12) to their respective groups (F(8,54) = 25, P < 0.0001; Wilk’s lambda = 0.04, a test of appropriateness of classifiers in predicting group membership of predictors), using the δ13CEAA values of the EAAs, isoleucine, phenylalanine, valine and lysine (Fig. 2). Additionally, two test fungus (Fusarium solani) samples were correctly classified as fungi. The classification of these fungal samples further validated the appropriateness of the model using the selected EAAs (Fig. 2). Wood was not used in the training data (Table S1), because we were interested in determining whether it would be correctly classified with the plant classifier. The posterior probabilities associated with the model classifications are summarized for both the model classifiers and the termite and wood samples in Table S2.

Figure 2 First discriminant analysis of termite, bacteria, fungi and plant samples.

Predictive modeling (LDA) using δ13CEAA data based on three classifier groups (plants (n = 12), fungi (n = 9), and bacteria (n = 11)) and three predictor groups (termite carcass (n = 5), termite gut filtrate (n = 5), and wood diet (n = 3)) using the EAAs; isoleucine (Ile), lysine (Lys), phenylalanine (Phe), and valine (Val). Wilks’ lambda = 0.09, P < 0.0001; LD1 = 92.6%, LD2 = 7.4%. 95% confidence limits decision regions for each group/classifier are depicted as ellipses around the classifiers and the decision boundaries between the groups/classifiers as lines.

Two wood samples fell within the 95% confidence limit decision region of the fungal classifier. This is suggestive of possible fungal infestation of the wood materials. The remaining third wood sample was in-between the fungal and bacterial classifier decision boundaries. Four termite carcass samples and two termite gut filtrate samples had discriminant scores within the 95% confidence limit decision region of the bacteria classifier, suggestive of possible bacterial EAA input (Fig. 2). One termite carcass sample was within the 95% confidence limit decision region of the fungal classifier, suggestive of fungal EAA input in that sample (Fig. 2). Three termite gut filtrate samples were outside of the 95% confidence limit decision region of the bacterial classifier, but were within the decision boundary of the bacterial classifier. This is taken to suggest likely bacterial EAA input in these samples (Fig. 2). The displacement of these termite gut filtrate samples is attributed to their 13C-depleted isoleucine values (Table S1).

The performance of the classification model without isoleucine from all samples was investigated, due to concerns about the influence of isoleucine on the displacement of samples in the LDA plot (Fig. 1). Bacteria, fungi and plant samples were correctly classified into distinct groups (F(8,54) = 28.9, P < 0.0001; Wilk’s lambda = 0.06, a test of appropriateness of classifiers in predicting group membership of predictors) (Table S3). As with the previous analysis, the test fungus, F. solani samples were similarly correctly classified as fungal in origin (Fig. 3), further validating the model and the classification in the absence of isoleucine. Omitting isoleucine from the model resulted in the placement of four termite gut filtrate samples within the 95% confidence limit decision region of the bacterial classifier, and the fifth one within the fungal classifier decision region (Fig. 3). Four termite carcass samples were classified as bacterial in origin, and three were located within the decision region of the bacterial classifier group. The fifth termite carcass sample was classified as fungal (Fig. 3). Classification of both termite carcass and termite gut filtrate samples in both models (Figs. 2 and 3) was essentially similar (Table S3). Omitting isoleucine in the second analysis, nonetheless, minimized the variance between samples and reduced the skewing of the samples within the LDA plot (Fig. 3). Based on the 13C-offset data and the results from both predictive model analyses, the hypothesis of gut microbial EAA is tentatively substantiated. Nonetheless, the sourcing of EAAs from extracellular wood-degrading fungi by termites in this study remains an additional/alternate possibility.

Figure 3 Second discriminant analysis of termite, bacteria, fungi and plant samples.

Predictive modeling (LDA) using δ13CEAA data based on three classifier groups (plants (n = 12), fungi (n = 9), and bacteria (n = 11)) and three predictor groups (termite carcass (n = 5), termite gut filtrate (n = 5), and wood diet (n = 3)) using the EAAs; lysine (Lys), phenylalanine (Phe), and valine (Val). Wilks’ lambda = 0.09, P < 0.0001; LD1 = 95.3%, LD2 = 4.6%. 95% confidence limits decision regions for each group/classifier are depicted as ellipses around the classifiers and the decision boundaries between the groups/classifiers as lines.

Discussion

The membership of the gut microbiome of wood-feeding termites like R. flavipes is varied and members perform several key important functions related to the host’s nutritional ecology. Essential amino acid provisioning by these gut microbes have been proposed, but remain to be empirically determined. In this study, we sought to determine gut microbial EAA provisioning to termite hosts by taking advantage of the natural variations in the δ13CEAA stable isotope signatures between plants, bacteria, and fungi.

A primary premise of this approach is that termite δ13CEAA would closely resemble dietary δ13CEAA (wood) in the absence of microbial provisioning. While we did not determine a significant difference between termite carcass and dietary δ13CEAA(Δδ13C = 1) (Table 1), there were notable variations in the 13C-offset patterns of isoleucine, leucine, and lysine, between termites and the wood diet (Fig. 1). Most wood-feeding lower termites are notably 13C-depletd compared to fungus feeding and soil-feeding termites (Tayasu, 1998). Degradation of cellulose by host and microbe-derived cellulolytic processes liberates carbon from wood in the termite gut (Watanabe & Tokuda, 2010). However, there are multiple microbial processes taking place in the termite gut besides cellulose degradation, including reductive acetogenesis (Brune & Friedrich, 2000), which affects the 13C-signatures of carbon liberated due to cellulose degradation and the carbon finally incorporated into insect host tissues. The higher incidences of microbial acetogenesis relative to methanogenesis in wood-feeding lower termites, followed by absorption of the newly formed and 13C-depleted acetate, is proposed to be responsible for the determined negative 13C-discrimination between wood-feeding lower termites and their woody diets (Bignell et al., 1995; Tayasu, 1998). While the Δδ13CEAA between termite carcass and diet was not greater than the posited 1‰, δ13C-depletion has been previously reported between termite and their wood diet based on bulk δ13C data (Bignell et al., 1995; Tayasu, 1998). Thus for termites, based on the diverse composition and functionalities of the gut microbiota, perhaps a 13C-offset of 1‰ may be sufficient to indicate non-dietary EAA input. Additionally, the individual variations in the 13C-offsets across all EAAs between the wood diet and termite carcass and gut filtrate samples, possibly further suggests non-dietary EAA input despite the overall 13C-offset of 1‰ (Fig. 1).

An investigation of the biosynthetic origins of EAAs in termite carcass and gut filtrate samples using the predictive model classified a majority of termite samples as bacterial in origin (Figs. 2 and 3), and diet (wood) samples as mainly fungal. Analyzing the data, with and without isoleucine, demonstrated the suitability of the EAAs used to adequately separate fungi, bacteria and plants in the predictive models. Unfortunately, not all EAAs passed quality control, and were therefore not available for use in further analyses. Of the 6 EAAs, measured from samples in this study, only 4 (Fig. 2) and 3 (Fig. 3) were used in the LDA. This is attributed partially to the difficulties inherent in the successful derivatization and analytical processes associated with quantifying δ13CEAA data. Increasing the number of EAAs used in the model, would essentially offer greater resolution as well as increased confidence in the subsequent model classifications.

Despite the limitation regarding the number and kinds of EAAs used, results from both predictive models show moderate support for the assertion of gut microbial sources of EAAs to the termite host (Fig. 2). These results however, are not definitive and require further validation. The major reasons for this are the absence of significant 13C-offsets between termite samples and three of the EAAs in the study (lysine, valine and phenylalanine), and the classification of diet (wood samples) as fungal in the predictive model (Fig. 2, Table S2). This suggests, perhaps, that the wood diet was compromised and was not appropriate for this effort. Fungal growth has been observed to quickly overtake laboratory termite colonies when populations are either small or stressed (S Jones, pers. comm., 2015). The classification of only one of the termite carcass samples as fungal is of importance, but does not detract from the bacterial origin of EAAs in the other four termite carcass samples (Figs. 2 and 3).

The relative significance/importance of fungal EAA input compared to bacterial EAA input was not the objective of this study. Fungal origins of EAA, however, cannot be ruled out, since fungi associated with termites may begin the cellulolytic process prior to termite ingestion, thus enhancing cellulose degradation (Hyodo et al., 2003). Ingested fungi may in turn be consumed and digested for EAAs, evidenced from the predominantly fungal EAA signature in one of the termite carcass samples. This scenario is not entirely unlikely, since fungal origins of EAA (as a component of gut microbial EAA input) have been previously documented in other insects such as, the solitary, wood-feeding Anoplophora glabripennis (Ayayee et al., in press).

Based on the social structure of R. flavipes colonies, and proctodeal trophallaxis between colony mates, (Nalepa, Bignell & Bandi, 2001; Shimada et al., 2013), digestion and assimilation of microbial EAA following proctodeal transfers is most likely the route for the bacterial/microbial EAA input observed in this study. Alternatively, direct absorption of microbial of EAA from across the walls of the hindgut paunch in termites is likely, in a manner similar to the uptake of acetate produced by hindgut microbial residents (Breznak & Kane, 1990).

Protists associated with termites are known to aid synergistically in cellulose degradation (Scharf et al., 2011), and serve as hosts to both ecto- and endo- symbiotic bacteria within the gut lumen (Ohkuma, 2008). Little is known about the essential amino acid biosynthetic capabilities of protists in general, but it is likely, that being eukaryotes, they lack the machinery necessary for de novo EAA biosynthesis (Ginger et al., 2010), but are capable of utilizing EAAs as substrate for metabolism. It remains to be conclusively determined that termite gut-associated protists are incapable of EAA biosynthesis, and thus the possibility that termites may be acquiring protist-derived EAAs cannot be entirely ruled out. Termite gut-associated protists however, are known to be obligate hosts to a variety of ecto- and endo-symbiotic bacteria present in termite microbiomes (Ohkuma, 2008). The functions of these protist-associated bacteria include nitrogen fixation, reductive acetogenesis and methanogenesis, as well as limited cellulose degradation (Ohkuma, 2008). These associated bacteria are currently regarded as essential sources of metabolites such as EAAs for their protist hosts. The protists together with associated bacteria then, subsequently serve as sources of EAA for the termite hosts upon digestion.

Overall, this study demonstrates the applicability of the 13C-fingerprint approach in investigating biosynthetic origins of EAA in insect-microbe systems. Additional improvements to the experimental design (careful selection of experimental dietary materials), increased sample numbers, and optimized procedures/protocols for δ13CEAA data generation via isotope ratio mass spectrometry, are needed, in order for definitive establishment of the role of gut-associated microbes of termites as sources of EAA.

Conclusions

In summation, this study provides promising evidence in support of putative gut microbial (bacterial and fungal) EAA provisioning in termites. Though the results presented herein are not exhaustive, they serve as the baseline for further work, investigating microbial EAA provisioning functions in greater detail. Finally, the results presented provide a framework/approach to, investigating gut microbial EAA provisioning in similar, and other insect-microbe symbiotic system using δ13C stable isotope analysis.

Supplemental Information

Table S1 Summary raw calibrated 13C data

Sample source and all mean inter-lab calibrated essential amino acid values (δ13CEAA) used in this study. Means are based on 2 technical replicates for each biological sample and 3 technical replicates for fungi, bacteria, and plant reference samples.

Click here for additional data file.

Table S2 Validation of model and sample classifications for second discriminant analysis in Fig. 2

Posterior probabilities of the classifier samples (fungi, bacteria, and Plants) and experimental group samples used in the predictive model plot in Fig. 2. (Wilks’ lambda = 0.09, P < 0.0001).

Click here for additional data file.

Table S3 Validation of model and sample classifications for second discriminant analysis in Fig. 3

Click here for additional data file.

We would like to thank Drs. Agustin Munoz-Garcia, Pamela Sullivan and Bryan Carstens for providing feedback during the preparation of the manuscript. We would also like to thank Dr. Thomas Larsen for providing feedback on data analysis and performing the interlab calibration carried out on the reference bacterial, and fungal samples. This calibration was performed using unpublished data generated by analyzing some of the same representative microbial samples from Larsen et al. (2013), at the Stable Isotope Facility (SIF) at UC Davis, Davis, California, USA.

Additional Information and Declarations

Competing Interests

Author Contributions

Ethics

The authors declare there are no competing interests.

Paul A. Ayayee conceived and designed the experiments, performed the experiments, analyzed the data, wrote the paper, prepared figures and/or tables.

Susan C. Jones and Zakee L. Sabree contributed reagents/materials/analysis tools, wrote the paper, reviewed drafts of the paper.

The following information was supplied relating to ethical approvals (i.e., approving body and any reference numbers):

No animal rights were violated in the execution of this study and conditions were within the guidelines of the Ohio State University’s Office of Responsible Research Practices.

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
