# Peer review of "Can 13C stable isotope analysis uncover essential amino acid provisioning by termite-associated gut microbes?"

_PeerJ, doi:10.7717/peerj.1218_

## Round 0.1 · original submission · Major Revisions

Both expert reviewers provide extensive suggestions regarding improvement of data interpretation, as well as additional experimentation (or otherwise down-tuning some of the conclusions).

Reviewer 1 ·

Basic reporting

The present study performed stable carbon isotope analyses of five essential amino acids derived from termite carcass, gut filtrate, and their dietary wood, supporting essential amino acid (EAA) provisioning by the gut bacteria in termites. Although this is an interesting attempt and meets the standard of this journal, interpretation of the results need some more precautions.

Experimental design

A technical question is why they selected the five EAAs, three of which (i.e. Lys, Phe, and Val) did not show significant differences of offset between wood and carcass, whereas remaining one of two (i.e. Leu) was unavailable in three samples in terms of delta13C EAA. Moreover, a previous paper has suggested that Phe shows the poorest resolution among plants, fungi, and bacteria (Larsen et al. 2009). In this context, the other EAAs should have been employed in the present analyses to validate the results (more details are described in “Validity of the Findings”).

Validity of the findings

Given EAAs in wood were derived from fungi and in the gut fluid were derived from bacteria as Fig. 2 and Table S2 indicated, Table 1 suggested that delta13C of termite EAA (carcass) was more similar to fungi (wood) rather than bacteria (the gut fluid). Based on Fig. 1, Lys, Phe, and Val could not resolve the differences, while Ile and Leu of termite (carcass) did not show significant differences from fungi (wood). On the other hand, it was not indicated whether significant differences were present between the carcass and the gut fluid. According to Larsen et al. (2009), Ile, Leu, and Lys gave the best resolutions to show significant differences between normalized delta13C EAA values between bacteria and fungi. Thus, the results appear to indicate that EAAs of the termite tissue (carcass) were similar to the dietary EAAs (fungi on wood). Although some discussions were found on this issue, these results cannot exclude the possibility that the termites obtained EAAs from fungi inhabiting the dietary wood, without showing differences or similarities between the carcass and the gut fluid. Nevertheless, LDA grouped EAAs of the termite tissue (carcass) into bacterial (but one of the five carcass samples was classified as fungal). However, Leu was omitted from the analysis and three EAAs (Lys, Phe, and Val) revealed similar delta13C values among samples investigated, raising the possibility that this analysis was strongly influenced by Ile. It was not clear whether the result could be consistent if the analysis were done with other EAAs, and thus delta13C values of other EAAs need be taken into considerations in this study.

Another big question is whether the intestinal protists contribute to provisioning of EAA. The hindgut of this termite is full of the flagellated protists (parabasalids and oxymonads), and the microbial composition in the gut of R. flavipes should be explained in Introduction. However, EAA synthesis by these protists in termites has been poorly studied. According to the genome analysis of a parasitic parabasalid, Trichomonas vaginalis, suggested some possible pathways of EAA biosynthesis (Carlton et al. 2007. Science). Is it possible to exclude the contribution of the gut protists to EAA biosynthesis in the present study? This point should be considered.

Additional comments

Although the present study suggested the provisioning of EAAs by the gut bacteria, no experiment was made to demonstrate the possible contribution of proctodeal trophallaxis. Based on the present study, one cannot exclude the possibility that EAAs are absorbed across the hindgut wall. Although this reviewer agrees to the possible contribution of proctodeal trophallaxis to this process, it should not be emphasized without experimental evidence.

Minor remarks
L71 and L266:
The usage of lignocellulose was incorrect. This is a complex of cellulose, hemicellulose, and lignin, while lignin degradation (mineralization) and assimilation has not been firmly established in termites. The term "lignocellulose" should be removed from the sentences.


L312-313:
This sentence is not scientific.

L313-314:
One termite sample was classified as fungal (Fig. 2 and Table S2).

There were many typos on literature citations. Please check them thoroughly.
Here are some examples;
“et al.” was often missing (Scharf 2011, Tokuda 2014, Newsome 2011).
Osamu and Kitade (2004) should be Kitade (2004).
Watanabe (2010) should be Watanabe and Tokuda (2010).
The doi number for Larsen et al. 2009 was invalid.
Bibliography of Tartar et al. (2009) should be Biotechnol. Biofuels 2: 25.
etc.

Fig. 2:
It is probably better to explain the confidence range (95%) of the dotted lines in this caption, too.

·

Basic reporting

This article appears to generally adhere to the basic reporting demands of PeerJ.

English of the paper, which could be improved, as there are many small grammatical errors throughout the paper.

The Discussion has a paragraph on unpublished data on cockroaches, which appears to possibly have been relevant to include in the current manuscript. If this for some reason is not possible or relevant, I suggest the authors either remove the paragraph or elaborate to ensure that readers have more information on the relevant similarities/differences.

Experimental design

This article appears to adhere to the experimental design demands of PeerJ.

Validity of the findings

The validity of the findings appear robust, although admittedly a few more replicates would have been desirable, as they are rather low.

While the research question and the approach is interesting and good, I was a bit disappointed by the lack of significant findings of the paper. The manuscript title should perhaps be rephrased, so that readers do not get the impression that the paper provides evidence for positive results, when it in fact is a negative / inconclusive result.

Additional comments

While this is an interesting paper, I would have loved to see more! This is an area that deserves much more attention and a need for more insight. I would therefore recommend, even if I give the paper a positive review, that you consider expanding the efforts to shed more light on EAAs in the lower termites.

---

## Round 0.2 · Minor Revisions

As you will see, both reviewers found the manuscript much improved. Please consider the few remaining comments provided by one of the reviewers.

Reviewer 1 ·

Basic reporting

Basically, the manuscript has been revised appropriately. I have a few minor remarks given below.

Experimental design

My previous concerns have been addressed appropriately.

Validity of the findings

Basically, my previous concerns have been addressed appropriately. If the authors believe that the contribution of the intestinal protists to EAA synthesis can be ruled out based on your data, this point should be clarified in the text. For your future reference, I have a few comments on this issue. In the rebuttals and the manuscript text (e.g. L113-L114 and L364-L365), the authors postulate that protists are unable to synthesize EAAs de novo. This could be true, but it is also true that a very limited number of data have been available on the ability of EAA synthesis by protists (as the authors also discussed (L363-L364)). I am also aware that many protists are likely lacking this ability as described by Payne and Loomis (2006), and some literatures described that the ability of EAA synthesis was probably lost during evolution of protists to animals. However, as described in your reference paper (Ginger 2010), eukaryotes are now divided into six supergroups (almost equivalent to Kingdoms), each of which contains so-called protists. In my opinion, protists are too diverse and available data are too limited to conclude that protists cannot synthesize EAAs de novo.

All intestinal symbionts in termites belong to the supergroup Excavata. According to KEGG pathways based on genome information of a few excavates, a number of genes that could be involved in EAA synthesis are fairly variable among species. R. flavipes harbors more than ten species of protists, whose nitrogen metabolism has yet to be clarified. These protists could obtain EAAs by direct provisioning from the symbiotic bacteria or digestion of these bacteria in food vacuoles, but I am not sure whether it can be ruled out that protists obtain inorganic nitrogen wastes from the symbiotic bacteria to produce EAAs by protists themselves.

Additional comments

L90 and L92: Replace “Osamu & Kitade, 2004” with “Kitade 2004”. (Osamu is the given name, while Kitade is his family name).

Results. L199: Remove the first sentence “We did not pre-select … in this study”. This is redundant and fully understandable by reading the next sentences.

Reference:
L405 Bignell et al. (1995): I think this article is a book chapter in “Forests and Insects” published in 1997 by Chapman & Hall. Please check the bibliography.

L409: Replace “Fems” with “FEMS”.

L461: Replace “Osamu K” with “Kitade O”.

L482 Tokuda et al. (2014): Replace “Suboi Y” with “Tsuboi Y”.

L498 Watanabe and Tokuda (2010): Replace “Watanabe HGT” with “Watanabe H”.

·

Basic reporting

No further comments.

Experimental design

No further comments.

Validity of the findings

No further comments.

Additional comments

The authors have in my view made satisfactory edits to accommodate the critique from the first round of reviews. I therefore have no additional comments or suggestions for improvements of the manuscript.

---

## Round 0.3 · accepted · Accept

All suggestions have been addressed appropriately.